# Autonomous Tree-search Ability of Large Language Models

## Abstract

Large Language Models (LLMs) have excelled in remarkable reasoning capabilities with advanced prompting techniques (*e.g.,* Chain-of-Thought), but they fall short on tasks that require exploration, strategic foresight, and sequential decision-making. Recent works propose to utilize external programs (*e.g.,* Python codes) to define search logic, such that LLMs can perform *passive* tree search to solve more challenging reasoning tasks. Though impressive results have been achieved, there are several fundamental limitations of these approaches. First, passive tree searches are not efficient as they usually require multiple rounds of LLM API calls to solve one single problem. Moreover, passive search methods are not flexible since they need task-specific program designs. Then a natural question arises: can we maintain the tree-search capability of LLMs without the aid of external programs, and can still generate responses that clearly demonstrate the process of a tree-structure search? To this end, we propose a new concept called *autonomous tree-search ability* of LLM, which can automatically generate a response containing search trajectories for the correct answer. Concretely, we first perform both BFS and DFS style search trajectories using more capable LLM API (*e.g.* GPT-4 and GPT-3.5) via a fixed system prompt, allowing them to perform autonomous tree-search (ATS) right out of the box. Experiments on 4 challenge puzzle games demonstrate our method can achieve huge improvements. The ATS-BFS method outperforms the Chain of Thought approach by achieving an average accuracy improvement of 33%. Compared to Tree of Thoughts, it requires 65.6% or 47.7% less GPT-api cost to attain a comparable level of accuracy. Moreover, we have collected a dataset using the ATS prompt method and fine-tuned LLaMA with this dataset. This approach has shown to yield a greater improvement compared to the ones fine-tuned on CoT data. Specifically, it outperforms CoT-tuned LLaMAs by an average of 40.6% and 38.5% for LLaMA2-7B and LLaMA2-13B, respectively.

## 1 Introduction

Large language models (LLMs) (*e.g.,* LLaMA (Touvron et al., 2023), GPT-3 (Brown et al., 2020), GPT-4 (OpenAI, 2023)) have demonstrated an increasing capability to perform a broader spectrum of reasoning tasks that involve math (Cobbe et al., 2021), logic (Liu et al., 2023), and algorithm execution (Jojic et al., 2023).

With more advanced prompting techniques, the reasoning ability of LLMs can be further improved. For example, the Chain-of-Thought (CoT) approach (Wei et al., 2022) lets LLMs perform step-by-step reasoning and achieve strong performances on several reasoning tasks. Some works tried to improve CoT's reasoning ability. For example, Hao et al. (2023) uses a world model to give intermediate world states to help reasoning, Gao et al. (2023) lets LLM output programs during reasoning. More recently, Yao et al. (2023) found that CoT is confined to left-to-right decision-making processes during inference, so it cannot handle more challenging reasoning tasks necessitating exploration, strategic foresight, and sequential decision-making. Several works (Yao et al., 2023; Long, 2023; Besta et al., 2023; Ye et al., 2023) employed hand-crafted search algorithms that determined the search logic and utilized LLMs to perform the left functions. We refer to such methods as *passive search*, since through these methods, LLMs perform as passive students guided by teachers who keep asking single-functional questions. The style of passive search has crucial drawbacks. First, passive search methods require multiple rounds of LLM API calls to solve a single problem.

This inevitably results in substantial financial costs (Chen et al., 2023) and an increase in carbon footprint (Wu et al., 2022; Dhar, 2020). Further, passive search methods are not flexible since they need task-specific program designs. This is significantly less convenient than interacting with chatbots possessing the CoT capability. When utilizing CoT, we can directly request a step-by-step response from LLMs in a chat scenario by simply asking, "Please provide a step-by-step answer." On the contrary, when utilizing passive search like Tree of Thoughts (ToT), we need to design the format of states, design the prompt for LLMs to list the next states or evaluate states, and design how to extract information from the response of LLMs. These are all required specific designs for each task.

To address the challenges in passive search, this work focuses on the "active" search and studies the so-called *autonomous tree-search ability*. That is, we let the large language model write down the tree-search process entirely by itself. With autonomous tree-search (ATS) ability, LLMs, without the aid of external programs, can generate responses that clearly demonstrate the process of a tree-structure search. LLMs with ATS ability can exhibit satisfactory flexibility, capability, and efficiency. Specifically, compared to CoT, ATS can explore a large set of possible solutions through tree search until it reaches a satisfying answer, while CoT only has one shot to produce the answer. Compared to ToT, ATS is self-reliant, costing only a single response from the LLMs without external assistance. We implement ATS in both ATS-BFS and ATS-DFS, defined by whether the trajectories are shown in the text of form BFS structure or DFS structure. (§ 3)

To examine the search ability of LLM, we set up four moderately challenging puzzles as evaluation datasets. Our experiment results show that with a carefully designed system prompt, GPT-4 could conduct ATS on all these puzzles and significantly improve its performance. (§ 4) Also, smaller models (*i.e.*, LLaMA 2 7B and 13B (Touvron et al., 2023) ) can be equipped with ATS through fine-tuning. We gathered the data produced by the ATS-enhanced GPT-4. Upon fine-tuning this data, the ATS-tuned LLaMAs demonstrated satisfactory search performance. They surpassed the CoT-tuned LLaMAs by an average of 40.6% and 38.5% for LLaMA2-7B and LLaMA2-13B, respectively. Moreover, when compared to ToT-tuned LLaMAs that involve some search capability, the ATS-tuned LLaMAs still exhibited better performance. (§ 5)

## 2 RELATED WORK

**Large Language Model.** It is commonly held that scaling up pretrained large models (PLMs) often yields enhanced performance for downstream tasks, an observation encapsulated by the so-called "scaling law" (Kaplan et al., 2020). Multiple research endeavors have probed the boundaries of performance by training increasingly vast PLMs, including the 175-billion-parameter GPT-3 (Brown et al., 2020) and the colossal 540-billion-parameter PaLM (Chowdhery et al., 2022). Nowadays, GPT-4 is the state-of-the-art (OpenAI, 2023). While GPT-4's achievements are significant, its opaque training processes and undisclosed architecture have hindered further open-source progress and in-depth research within this field. Offering a refreshing contrast, Vicuna (Platzer & Puschner, 2021), LLaMA (Touvron et al., 2023) and Alpaca (Taori et al., 2023), breaks through these barriers as a transparent, open-source chatbot equipped with an expanded dataset and a user-friendly, scalable infrastructure. In the rapidly evolving landscape of Language Model-based chatbots, LLaMA 2 (Touvron et al., 2023) has assertively carved out a prominent position as a leader in the field.

**Reasoning.** In recent developments, there has been notable progress in Large Language Models (LLMs), especially regarding their emergent properties and context-specific learning capabilities. Such advancements pave the way for new horizons in machine reasoning (Wei et al., 2022). By utilizing chain-of-thought prompts (Wei et al., 2022) and various cues, researchers have demonstrated that these models can systematically solve mathematical and logical reasoning tasks (Kojima et al., 2022; Drori et al., 2022). Building on this foundation, recent research has ventured into generating multiple solutions, subsequently leveraging self-consistency (Wang et al., 2022) to ascertain the most appropriate response. Furthermore, to enhance performance on exploration-related questions.

**Search Capability of LLM.** In order to achieve passive search capabilities, previous research (Yao et al., 2023; Long, 2023; Besta et al., 2023; Ye et al., 2023) has employed hand-crafted search program to dictate the search logic, while leveraging LLMs to provide heuristic guidance. More specifically, Yao et al. (2023); Long (2023) utilized a human-programmed approach to execute Breadth

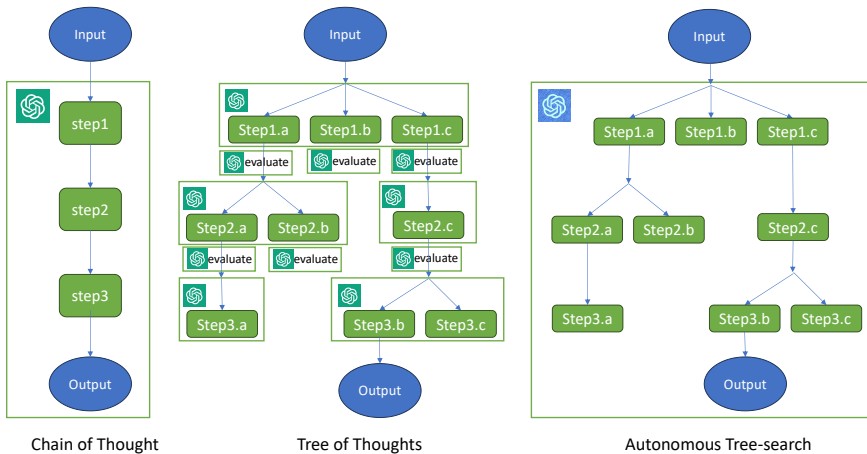

(a) Chain of Thought vs. Tree of Thoughts vs. Autonomous Tree-search

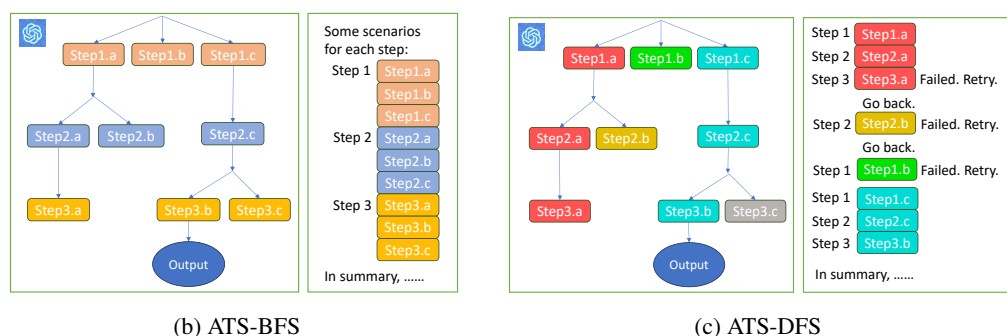

(b) ATS-BFS          (c) ATS-DFS

Figure 1: These figures provide an overview of Autonomous Tree-search (ATS) in comparison with CoT and ToT, while also illustrating the process by which the tree structure is flattened into a text paragraph. In Figure (a), a solid block within a step symbolizes a thought, and a box signifies a single message chat with LLMs. Figures (b) and (c) offer additional clarification on ATS, detailing each case of ATS-BFS and ATS-DFS.

First Search (BFS) and Depth First Search (DFS), incorporating an internal LLM-assisted function. Besta et al. (2023) expanded the tree structure to a graph structure, while Ye et al. (2023) harnessed an Elo-based Self-Judgment Mechanism for decision-making. Zhang et al. (2023) provided a human-designed space of search logic instead of a specific search logic and ask LLMs determine which logic to preform. However, these methods necessitate the support of code logic. As discussed before, passive search methods are not only costly but also incapable of providing direct assistance via chat, and necessitate task-specific design. As for autonomous search, a concurrent research proposed Algorithm of Thoughts (Sel et al., 2023). The primary objective of their approach is to significantly reduce the number of queries employed by existing multi-query reasoning methods, while maintaining performance for tasks that necessitate adept application of world knowledge. This is intended to foster a more efficient and responsible utilization of AI resources. However, this study was limited to GPT-4 and few-shot in-context learning. In comparison, our work additionally considers smaller language models and zero-shot settings, giving a more comprehensive investigation towards Autonomous Tree-Search (ATS) ability.

**Control and Enhance LLM Behaviour.** Large-scale models like GPT-3 and GPT-4 have shown impressive performance across a range of tasks through in-context learning. This has led to the widespread belief that prompts can be used to impart new knowledge to GPT and alter its behavior. For instance, it has been demonstrated that algorithmic reasoning ability can be taught through in-context learning (Zhou et al., 2022). In the case of smaller models, it is common practice to finetune them to excel in specific tasks (Rajani et al., 2019; Talmor et al., 2018; Hendrycks et al., 2021; Nye et al., 2021).

# 3 AUTONOMOUS TREE-SEARCH METHOD

## 3.1 FRAMEWORK

Figure 1 provides a concise overview of Autonomous Tree-search, illustrating that the response generated by Large Language Models (LLMs) encompasses tree-structured search trajectories. There are two main methods for flattening tree-structures into text. If the trajectories have Breadth-First Search (BFS) structure, we refer to the method as ATS-BFS, and if they have Depth-First Search (DFS) structure, we call it ATS-DFS.

**ATS-BFS.** This approach, as depicted in Figure 1b, is grounded in BFS. When a problem is presented to LLMs, they explore from the shallow to deeper levels, iteratively maintaining various scenarios after each step. In other words, at each step, there are certain scenarios preserved in the previously generated text, and then LLMs continuously generate successors of these scenarios for the next step. This iterative process concludes when a solution is discovered. Compared to DFS-like exploration, BFS-like exploration is logically simpler. We will later demonstrate the effectiveness of BFS-like exploration in the GPT-4 experiment using a straightforward global fixed system message.

**ATS-DFS.** This approach, as shown in Figure 1c, is based on DFS. When a problem is presented to LLMs, they immediately attempt a solution, retreating and initiating another attempt from the current position. As DFS-like exploration necessitates complex logic, it performs optimally in the few-shot setting of GPT-4 and is less suitable for smaller models.

The detailed methodologies to enhance LLMs with ATS ability are different for large models and small models. They are further discussed in § 4.1 and § 5.1.

## 3.2 DISCUSSION

Compared to the Chain of Thought (CoT), LLMs with ATS ability can explore a significantly larger number of scenarios through its tree structure, while CoT is limited to a single trajectory. In contrast to the Tree of Thought (ToT), ATS relies solely on the LLM itself, requiring only a single response without external assistance. ToT, on the other hand, is heavily dependent on external code logic, necessitating multiple chat messages for numerous smaller steps. We will now discuss the flexibility, capability, and efficiency of ATS in comparison to these baselines.

**Flexibility**. ATS, like CoT, is an LLM behavior that does not require human intervention. It can be incorporated into LLMs through prompting or supervised training. In contrast, ToT requires a specific human-designed program for each specific task.

**Capability**. ATS can navigate numerous scenarios in a tree structure, similar to ToT. This capability allows ATS to significantly outperform CoT when tasks necessitate search. Moreover, ATS has a comprehensive view of the entire tree structure, enabling it to gather information from other branches. This advantage allows ATS to outperform ToT in certain instances.

**Efficiency**. As illustrated in Figure 1a, both CoT and ATS require only a single message call on LLMs, whereas ToT necessitates a large number of message calls. Since each method requires messages of varying lengths, in the following section, we will use the GPT-4 cost as a metric to evaluate efficiency. Compared to ToT, ATS is a significantly more efficient method.

# 4 ENHANCE GPT-4 THROUGH PROMPT

## 4.1 PROMPT METHODOLOGY

GPT-4 shows a remarkable ability to adhere to instructions. It is a common practice within the AI community to utilize system messages to direct GPT-4 to "role-play" a specific character. For example, if GPT-4 is instructed to mimic a Socratic-style teacher who provides hints rather than direct answers, it will faithfully assume this role, refraining from giving explicit answers.

Consequently, an efficient approach to teaching GPT-4 with ATS ability is through system messages. We supply GPT-4 with a uniform system message (Appendix D) across all tasks. This message briefly introduced ATS ability and encouraged GPT-4 to role-play an assistant who is good at ATS.

In this context, we refer to the approach through system messages as the zero-shot method, given that there is no task-specific prompt. Furthermore, we will explore the model's performance in a few-shot setting, providing some task-specific examples that execute ATS.

## 4.2 DATASETS

The datasets consist of four puzzles. These puzzles are derived from daily scenarios such as the Drop Water Puzzle that faces the situation of two unmarked water cups in daily life. The puzzles include the Drop Water Puzzle, Number Path Puzzle, Arithmetic Puzzle, and Minimal Grass Puzzle. The first three are solution-finding puzzles, ensuring the existence of a solution, with the answer defining the parameters of the solution. The last puzzle is an optimization puzzle. Further details can be found in Appendix B.

**Drop Water Puzzle.** Given four integers $a$, $b$, $c$, and $n$. You are given two empty bottles without scales of capacities of $a$ and $b$ liters and a large water reservoir. The goal is to get exactly $c$ liters of water within $n$ operations.

**Number Path Puzzle.** Given three integers $n$, $a$, and $b$ where $a < b$. Create a sequence of exactly $n$ mathematical operations, starting from the number $a$ and ending at $b$, using only the operations of doubling or increasing by one.

**Arithmetic Puzzle.** Given four integers $a$, $b$, $c$, and $n$. Use the numbers $a$, $b$, and $c$ and arithmetic operations to achieve a final result of $n$.

**Minimal Grass Puzzle.** Given three integers $a$, $b$, and $c$. Figure out the dimensions of three rectangular buildings with given floor areas, ensuring they don't block each other's view, and then arrange them to minimize the surrounding green space. The dimensions must be integer values.

Our datasets were created due to the limited research on search ability and the small number of puzzles used in existing studies, such as Tree of Thoughts (ToT), which only uses three puzzles which may not be suitable as they require additional skills. For example, ToT's Word Puzzle tests the vertical comprehension ability of LLMs, making it unsuitable for studying the search capability of LLMs, especially in our research with smaller models. On the contrary, the puzzles in our work focus better on search ability.

## 4.3 EXPERIMENT CONFIGURATIONS

In addition to our **ATS-BFS** and **ATS-DFS** methods, we also use **CoT** and **ToT** as baseline methods and implement all of the methods for both zero-shot and few-shot settings:

- **ATS-BFS** and **ATS-DFS.** In the zero-shot setting, we use fixed system messages for ATS-BFS and ATS-DFS respectively. The messages can be found in Appendix D.1 and Appendix D.2. The messages do not contain task information, so it is indeed a zero-shot setting. In the few-shot setting, we designed four examples. The example not only contains task-specific information but also shows how to perform ATS-BFS or ATS-DFS. Hence there is no system message in the few-shot setting.

- **Chain of Thought (CoT).** Given that the GPT-4 model already possesses CoT capability, we utilize GPT-4 with the "think step by step" prompt as the CoT method. We also designed four example instances for the few-shot setting.

- **Tree of Thought (ToT).** We implement the ToT algorithm as described in Yao et al. (2023). The process involves: 1) using GPT-4 to generate the next possible states from all current candidate states, 2) evaluating all potential next states, and 3) retaining a select number of top-rated states, denoted as "width". Although ToT is typically implemented in a few-shot setting, we also apply it in a zero-shot setting. This involves designing a detailed prompt for each function (*i.e.*, proposer, evaluator), and instructing GPT-4 on the rules and its current role.

Furthermore, as the performance and cost of ToT are sensitive to its width, and all other methods can enhance performance by increasing cost through self-consistency (Wang et al., 2022) (i.e., repeat and select the best), we establish two settings for all methods: a low-cost setting and a high-cost setting.

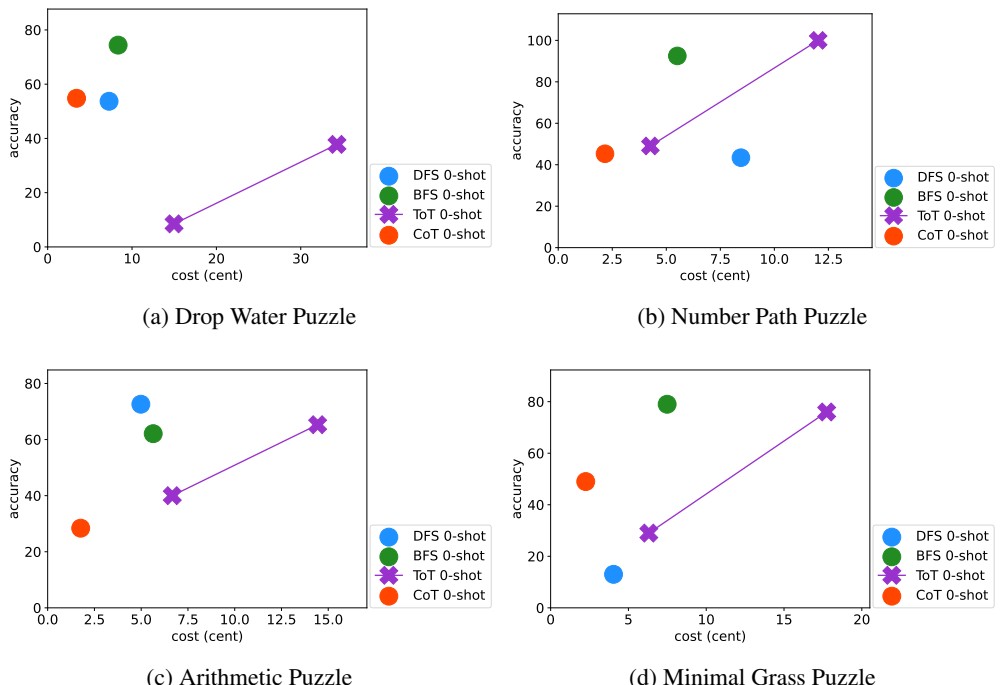

Figure 2: This figure illustrates the performance of GPT-4 in a zero-shot setting. The x-axis denotes the cost associated with the GPT-4 API, while the y-axis signifies the accuracy. Consequently, points that are positioned higher and more to the left depict the most effective and efficient results. The outcomes of ToT are in both low-cost and high-cost settings which are displayed as a polyline, while other results are shown in a low-cost setting.

- **Low-cost setting.** For ToT, the width is set to 1. For other experimental configurations, nothing special, call the message once without self-consistency.
- **High-cost setting.** For ToT, the width is set to 5. For other experimental configurations, they do not have the attribution "width", so we apply a self-consistency value of 3.

## 4.4 EXPERIMENT RESULT

We test accuracy for each setting, as well as the input tokens usage and output tokens usage. For ToT, we sum up the usage over all rounds of chat. The cost is estimated at the time of writing. (Input: 0.03/1K tokens; Output: 0.06 / 1K tokens;) The full table is shown in Appendix A.

**Zero-shot.** Figure 2 illustrates the results in a zero-shot setting across the four puzzles. Upon comparing performance and cost, we observe that 1) CoT consistently incurs the least cost, but its performance significantly lags behind the best. 2) ToT, with its expanded width, can achieve high accuracy at a high cost, but it performs poorly in a low-cost setting. Furthermore, ToT sometimes exhibits subpar performance, as it heavily relies on state evaluation, and the states in the Drop Water Puzzle are challenging for it to evaluate in a zero-shot setting. 3) Generally, ATS-BFS is the most effective method with a moderate cost. Specifically, ATS-BFS is at a cost level comparable to ToT (width=1), and it significantly outperforms CoT and ToT (width=1). Moreover, ATS-BFS displays comparable or better performance to ToT (width=5) at a much lower cost. 4) As for ATS-DFS, it demonstrates inconsistent performance. While ATS-DFS can achieve dominant performance in some cases (i.e., Arithmetic Puzzle), it sometimes fails to match the performance of CoT. We attribute this to the fact that DFS capability is not well generalized in LLMs, as evidenced by comparing zero-shot and few-shot settings. DFS logic is complex and often requires task-specific design.

**Few-shot.** Figure 3 presents the results in a few-shot setting across the four puzzles. Upon comparing performance and cost, we observe that 1) CoT consistently incurs the least cost, but its performance significantly lags behind the best. 2) ToT, with its expanded width, can achieve high accuracy at a high cost, and it demonstrates decent performance even in a low-cost setting, but fails some-

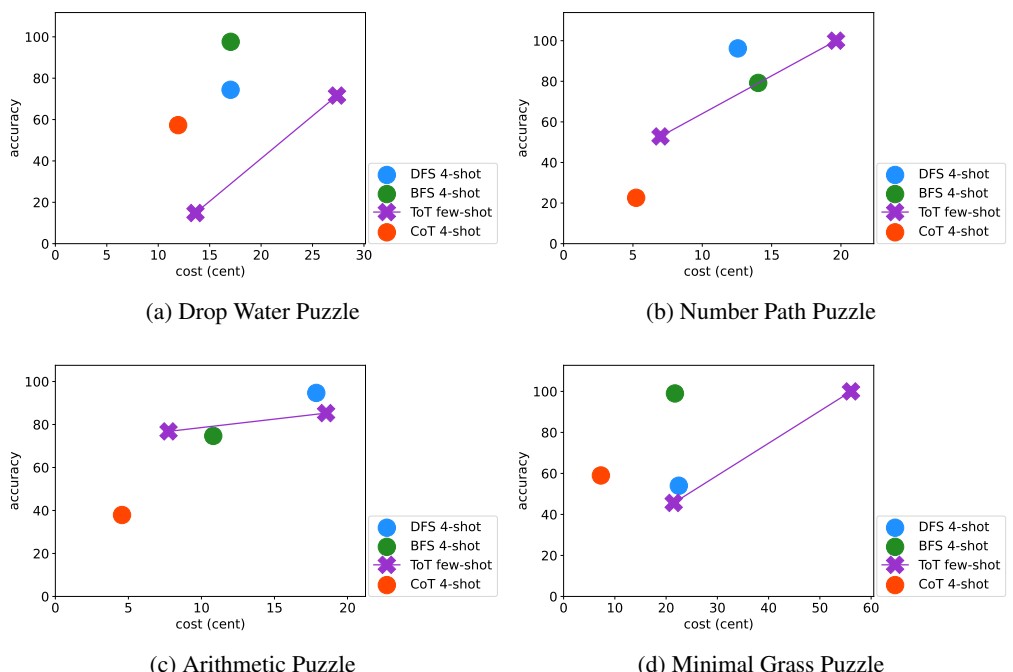

Figure 3: This figure illustrates the performance of GPT-4 in a few-shot setting. The x-axis denotes the cost associated with the GPT-4 API, while the y-axis signifies the accuracy. Consequently, points that are positioned higher and more to the left depict the most effective and efficient results. The outcomes of ToT are in both low-cost and high-cost settings which are displayed as a polyline, while other results are shown in a low-cost setting.

times (Drop Water Puzzle). 3) Although ATS-BFS significantly outperforms CoT, it only exhibits comparable or better performance to ToT. ATS-BFS benefits less from the few-shot setting than the zero-shot setting. 4) As for ATS-DFS, it emerges as the generally best method in solution-finding puzzles. Only in the optimization puzzle (Minimal Grass Puzzle), ATS-DFS performs poorly.

## 4.5 ADDITIONAL EXPERIMENTS

### 4.5.1 REAL COMPLEX REASONING TASK

A realistic and challenging reasoning task: CrossWords, one of the tasks in Tree-of-Thoughts. Table 1 shows the result, indicating:

- ATS handles real complex reasoning tasks better than CoT.
- All in-context methods have similar orders of magnitude of cost, while ToT incurs much higher costs.
- One major limitation of ATS is its constraint on the ability of LLMs. This task requires a strong understanding of rows and columns, and GPT-4 often fails in this aspect. Conversely, ToT decomposes some of the difficulty of rows and columns through Python code. (*e.g.*, writing back to the table with a row/column, extracting a row/column to a flat style before evaluation)

### 4.5.2 COMBINE ToT AND ATS

We also shows another application of ATS is to incorporate ATS into State Evaluator of ToT. This also emphasize that ATS and ToT are not in conflict.

Table 2 the results of two tasks are shown below, indicating ATS technique can enhance ToT by strengthening evaluator.

Table 1: Crosswords: a task requiring both search ability and spatial understanding ability

|  | Letter | Words | Game | input | output |
|---|---|---|---|---|---|
| IO (few-shot) | 38.7 | 14 | 0 | 790.25 | 30.505 |
| CoT (few-shot) | 40.6 | 15.6 | 1 | 1448.25 | 162.81 |
| ToT (few-shot) | 78 | 60 | 20 | >584306.45 | >848.05 |
| ATS-BFS (one-shot) | 46.6 | 18.5 | 0 | 1549.25 | 1211.75 |

Table 2: Combine ToT and ATS

|  | Drop Water | Number Path |
|---|---|---|
| ToT $_{width=1}$ **without** ATS-evaluator (few-shot) | 14.8 | 52.8 |
| ToT $_{width=1}$ **with** ATS-evaluator (few-shot) | 73.2 | 90.6 |

# 5 ENHANCE SMALL MODELS BY FINE-TUNING

## 5.1 FINE-TUNE METHODOLOGY

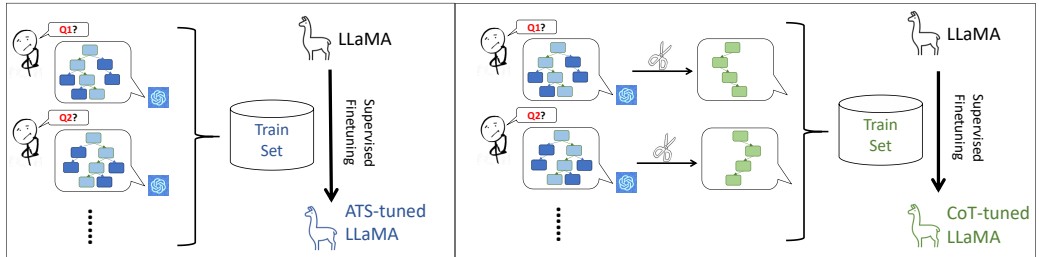

Figure 4: A brief view of finetuning. The left part is ATS-tuned LLaMA pipeline, while the right part is CoT-tuned LLaMA pipeline. The difference is whether to prune the tree structure or not.

In comparison to GPT-4, numerous smaller LLMs also have a substantial impact in daily life due to their cost-effectiveness and convenience. Consequently, we employ supervised fine-tuning, distilling from ATS-enhanced GPT-4 or from the messages generated by ToT method to smaller LLaMA models, to illustrate the advantages of our approach for smaller models.

As depicted in the left part of Figure 4, we gather text generated by the GPT-4 with ATS ability for a specific task. We then use these question-answer pairs to serve as supervised data for fine-tuning the LLM. In our experiment, we utilized LLaMA 2, culminating in an ATS-tuned LLaMA.

We can also prune the tree structure to extract the final solution in a chain structure as supervised data for another experiment setting. The right section of Figure 4 illustrates this. This pruning ensures that the text comprises only a chain rather than a tree, resulting in a Chain-tuned LLaMA.

More finetune settings are discussed in next subsection.

## 5.2 EXPERIMENT CONFIGURATIONS

We use LLaMA 2 as the base model. The process of obtaining training data involves initially acquiring raw data from either ATS or ToT (the raw data from ToT comprises multiple rounds of LLM messages). Subsequently, we convert this raw data into text, which takes on either text containing chain information or text containing tree information. With the data containing chain information, we call the LLaMA CoT-tuned, otherwise ATS-tuned or ToT-tuned.

- **Tuned Type.** There are three tuned types in total: ATS-tuned, ToT-tuned, and CoT-tuned. When applying ATS-tuned, we directly use the ATS output. In terms of ToT-tuning, we flatten the multiple rounds of messages into a single text for one data instance. When applying CoT-tuned, we extract the chain information from the raw data and rearrange it into a CoT response. ATS-tuned LLaMA gains ATS-ability, while CoT-tuned LLaMA gains CoT-ability.

- **Data Source.** This pertains to the method of raw data generation. Any erroneous data is eliminated from the raw data. To augment the quantity of usable raw data, we aim to generate it in as high-cost a setting as feasible. Specifically, ATS methods are executed with 5 self-consistency (selecting the best out of 5 trials), while ToT methods are run with a width of 5.

When flattening ToT raw data into text, the text length imposes a limitation. Given the abundance of information that needs to be flattened into text, such as potential successors and their evaluations, we have to restrict the width to 2 to accommodate all ToT information within the text length constraint. An example can be found in Appendix E.4.

## 5.3 EXPERIMENT RESULT

Table 3 presents the comprehensive results of the fine-tuned LLaMA 2.

The performance of CoT-tuned LLaMAs appears to be minimally influenced by the data source. This is likely due to the fact that most puzzles have a unique solution.

ATS-tuned LLaMAs achieve the highest performance. 1) Generally, ATS-BFS-sourced ATS-tuned LLaMA achieves the highest performance. 2) ATS-DFS-sourced ATS-tuned LLaMA performs exceptionally well in certain cases (e.g., Arithmetic Puzzle), but fails almost entirely in optimization tasks (e.g., Minimal Grass Puzzle).

For the ToT-tuned LLaMA, it is possible that some messages in the overall ToT process are complex and even redundant, making the structure more difficult for smaller LLMs to learn. As a result, it not only shows rare improvement upon ToT-tuned LLaMAs but also fails sometimes and exhibits lower than (e.g. Drop Water Puzzle, Minimal Grass Puzzle).

Table 3: LLaMA 2 Fine-tuned Result

| Model | Data Source | Tuned type | Performance (Accuracy) | | | |
|---|---|---|---|---|---|---|
| | | | DropWater | NumberPath | Arithmetic | MinimalGrass |
| LLaMA 2 7B | ATS-BFS | CoT-Tuned | 58.5 | 15.1 | 28.4 | 69.0 |
| | ToT$_{width=5}$ | CoT-tuned | 57.3 | 7.5 | 22.1 | 53.0 |
| | **ATS-BFS** | **ATS-tuned** | **74.4** | **94.3** | 51.6 | **88.0** |
| | **ATS-DFS** | **ATS-tuned** | 64.6 | 56.6 | **76.8** | 23.0 |
| | ToT$_{width=2}$ | ToT-tuned | 28.0 | 17.0 | 35.8 | 33.0 |
| LLaMA 2 13B | ATS-BFS | CoT-tuned | 59.8 | 20.8 | 33.7 | 69.0 |
| | ToT$_{width=5}$ | CoT-tuned | 58.5 | 26.4 | 24.2 | 54.0 |
| | **ATS-BFS** | **ATS-tuned** | **75.6** | **100** | 72.6 | **82.0** |
| | **ATS-DFS** | **ATS-tuned** | 64.6 | 62.3 | **85.3** | 15.0 |
| | ToT$_{width=2}$ | ToT-tuned | 25.6 | 41.5 | 44.2 | 25.0 |

## 6 CONCLUSION

This study presents a thorough examination of the Autonomous Tree-search (ATS) Ability of Large Language Models (LLMs) that enables them to excel in tasks requiring exploration with minimal queries. Our findings indicate that this ability can be activated through a fixed system prompt for large models or acquired through in-domain fine-tuning by small models. The ATS performance has demonstrated significant enhancements in accuracy, surpassing previous single-query techniques such as the Chain of Thought in various settings. Furthermore, ATS exhibits a better balance between performance and cost, particularly in zero-shot settings, compared to the earlier Tree of Thought passive search method. Future research includes training larger LLMs, specifically those exceeding 70B, with ATS ability. It remains uncertain whether a comprehensive performance degradation (*i.e.*, fee) occurs when acquiring the ATS capability, similar to the acquisition of instruction following ability, and whether the ATS capability could ultimately enhance other abilities akin to search ability.

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

## A    DETAILED MAIN RESULT

Table 4 shows the accuracy on four puzzles by Large Models (GPT-4) in different settings.

## B    DATASET DETAILS

Table 5 shows the size of datasets.

Table 4: Accuracy on four puzzles by Large Models (GPT-4) in different settings

| Task | Method | Low Cost Setting | | | High Cost Setting | | |
|---|---|---|---|---|---|---|---|
| | | accuracy | input | output | accuracy | input | output |
| Drop Water | CoT 0-shot | 54.8 | 472 | 334.94 | 76.8 | 472 | 1127.85 |
| | BFS 0-shot | **74.4** | 756 | 1014.11 | **81.7** | 756 | 2566.24 |
| | DFS 0-shot | 53.7 | 671 | 877.46 | 75.6 | 671 | 1614.29 |
| | ToT 0-shot | 8.5 | 4244 | 375 | 37.8 | 9096 | 1168 |
| | CoT few-shot | 57.3 | 3107 | 435.83 | 75.6 | 3107 | 850.17 |
| | BFS few-shot | **97.6** | 4883 | 399.24 | **100** | 4436 | 1210.12 |
| | DFS few-shot | 74.4 | 4173 | 752.32 | 89.0 | 3936 | 1563.78 |
| | ToT few-shot | 14.8 | 4072 | 234 | 71.6 | 8043 | 541 |
| Number Path | CoT 0-shot | 45.3 | 198 | 261.6 | 77.36 | 198 | 804.25 |
| | BFS 0-shot | **92.5** | 482 | 677.43 | 98.1 | 482 | 2040.89 |
| | DFS 0-shot | 43.4 | 397 | 1210.38 | 83.0 | 397 | 2893.96 |
| | ToT 0-shot | 49.1 | 1201 | 110 | **100.0** | 3376 | 317 |
| | CoT few-shot | 22.6 | 1263 | 240.53 | 49.1 | 1263 | 435.34 |
| | BFS few-shot | 79.2 | 3660 | 508.77 | 79.2 | 3213 | 1524.21 |
| | DFS few-shot | **96.2** | 3194 | 496.81 | 98.1 | 2957 | 1878.04 |
| | ToT few-shot | 52.8 | 1976 | 177 | **100.0** | 5553 | 498 |
| Arithmetic | CoT 0-shot | 28.4 | 182 | 203.37 | 37.9 | 182 | 573.22 |
| | BFS 0-shot | 62.1 | 466 | 707.46 | 81.0 | 466 | 1634.83 |
| | DFS 0-shot | **72.6** | 381 | 639.84 | **91.6** | 381 | 1469.4 |
| | ToT 0-shot | 40.0 | 1195 | 513 | 65.3 | 2639 | 1086 |
| | CoT few-shot | 37.9 | 1298 | 111.52 | 61.1 | 1298 | 384.39 |
| | BFS few-shot | 74.7 | 2907 | 348.15 | 85.3 | 2460 | 1020.84 |
| | DFS few-shot | **94.7** | 4666 | 644.05 | **96.8** | 4429 | 1950.67 |
| | ToT few-shot | 76.8 | 1755 | 414 | 85.3 | 4250 | 964 |
| Minimal Grass | CoT 0-shot | 49.0 | 320 | 216.07 | 58.0 | 320 | 499.0 |
| | BFS 0-shot | **79.0** | 604 | 946.17 | **92.0** | 604 | 2542.43 |
| | DFS 0-shot | 13.0 | 519 | 414.43 | 49.0 | 519 | 1511.85 |
| | ToT 0-shot | 29.0 | 1672 | 216 | 76.0 | 4736 | 589 |
| | CoT few-shot | 59.0 | 2212 | 105.96 | 64.0 | 2160 | 317.24 |
| | BFS few-shot | **99.0** | 5508 | 866.3 | **100.0** | 5009 | 2428.77 |
| | DFS few-shot | 54.0 | 5278 | 1107.68 | 70.0 | 5041 | 3067.03 |
| | ToT few-shot | 46.0 | 4170 | 1495 | **100.0** | 12423 | 3134 |

## B.1 DROP WATER PUZZLE

Given four integers $a, b, c, n$. There are two empty bottles with capacities of $a$ and $b$ liters and a large water reservoir. The goal is to get exactly $c$ liters of water in either bottle within $n$ steps, by either filling or emptying a bottle completely, or pouring water from one bottle to the other until one is full or the other is empty.

We prepare all possible cases for the puzzle, where the capacity of the two containers ranges between 5 and 30, and the number of steps is limited to 4 or fewer. We randomly select 82 cases to serve as the test set, while the remaining cases constitute the training set.

## B.2 NUMBER PATH PUZZLE

Number Path is a puzzle that involves finding a path between two numbers. In this puzzle, you are provided with a starting number and a target number. Your task is to transform the starting number into the target number in exactly four steps. For each transformation, you have the option to either double the current number (x2) or add one (+1) to it.

We restrict the start number to less or equal to 20, and the goal number to less or equal to 100. We collect all the pairs that can be reached with exactly 4 steps. There are 476 cases in total. We randomly select 95 of them as the test set, while the remaining as the train set.

### B.3 ARITHMETIC PUZZLE

Arithmetic Puzzle is a challenge that involves using arithmetic operations on initial numbers to reach a specified goal number. More specifically, you are given three numbers and a goal. Your task is to strategically use arithmetic operations to link these initial numbers in order to achieve the desired outcome.

We set the goal as one of 6, 8, 12, 16, 18, and 24. And we have imposed a restriction that all initial numbers must be less than or equal to 12. Under these conditions, we collect all possible problem scenarios, amounting to a total of 425 cases. We have randomly selected 53 cases to constitute the test set, with the remaining cases forming the training set.

### B.4 MINIMAL GRASS PUZZLE

Minimal Grass Puzzle involves determining the dimensions of three rectangular buildings, given their floor areas. The length and width of each building must be integers. The buildings must be positioned in such a way that they do not obstruct each other's view horizontally or vertically. The surrounding area in the bounding box of these buildings will be filled with green space. The objective is to minimize the area of this green space.

The given areas are randomly and uniformly selected from a range of 1 to 15.

Table 5: Dataset information

|                 | Drop Water | Number Path | Arithmetic | Minimal Grass |
|-----------------|------------|-------------|------------|---------------|
| train instances | 578        | 381         | 372        | 300           |
| test instances  | 82         | 95          | 53         | 100           |

## C EXPERIMENT HYPERPARAMETERS

- GPT-4 Experiment
  - CoT / ATS-BFS / ATS-DFS (best of 1 / no self-consistency): temperature = 0.2;
  - CoT / ATS-BFS / ATS-DFS (best of 3 / self-consistency=3): temperature = 0.7;
  - ToT: temperature = 0.7, evaluate voters = 3;
- LLaMA 2 Experiment
  - Training on one node with 8 GPUs.
  - Less than 3000 tokens per batch per GPU.
  - AdamW(lr=1e-5), update 250 iterations.
  - Generating use sampling with temperature = 0.2.

## D PROMPT

In this section, we show our system prompts of ATS-BFS and ATS-DFS for GPT-4.

### D.1 ATS-BFS ZERO-SHOT PROMPT

```
When you are solve a puzzle, if you can't ensure this step is the best
    for following steps, you should write down some possible scenarios to
     ensure a broad range of attempts. Here is an example of your
    response format:

Step 1
scenario 1, [initial state]-> (operation 1) [state 1]
scenario 2, [initial state]-> (operation 2) [state 2]
```

```
Step 2
scenario 1.1, [initial state]->[state 1]-> (operation 1) [state 1.1]
scenario 1.2, [initial state]->[state 1]-> (operation 2) [state 1.2]
scenario 2.1, [initial state]->[state 2]-> (operation 1) [state 2.1]
scenario 2.2, [initial state]->[state 2]-> (operation 2) [state 2.2]

Step 3
scenario 1.1.1, [initial state]->[state 1]->[state 1.1]-> (operation 1) [
    state 1.1.1]
(You should write around 8 lines for Step 3)

Step 4
scenario 1.1.1.1 ...
(You should write around 16 lines for Step 4)
...
```

## D.2  ATS-DFS ZERO-SHOT PROMPT

```
When you are solving a puzzle, if you find that a certain step cannot be
    successful, you should step back appropriately. Here is an example of
     your response format:

Step 1
[initial state]-> (operation) [state 1]

Step 2
[initial state]->[state 1]-> (operation) [state 2]

Step 3
[initial state]->[state 1]->[state 2]-> (operation) [state 3]

This is not the goal. Let's step back. Now it is [state 2].

Step 3 (revised)
[initial state]->[state 1]->[state 2]-> (operation) [state 3]

Let's step back. Now it is [state 2].

Let's step back. Now it is [state 1].

Step 2 (revised)
[initial state]->[state 1]-> (operation) [state 2]

...
```

## E  EXAMPLES

## E.1  GPT-4 ATS-BFS ZERO-SHOT OUTPUT

```
Step 1
scenario 1, [2, 3, 10]-> (2 + 3) [5, 10]
scenario 2, [2, 3, 10]-> (2 * 3) [6, 10]
scenario 3, [2, 3, 10]-> (2 + 10) [12, 3]
scenario 4, [2, 3, 10]-> (2 * 10) [20, 3]
scenario 5, [2, 3, 10]-> (3 + 10) [13, 2]
scenario 6, [2, 3, 10]-> (3 * 10) [30, 2]

Step 2
scenario 1.1, [2, 3, 10]->[5, 10]-> (5 + 10) [15]
```

```
scenario 1.2, [2, 3, 10]->[5, 10]-> (5 * 10) [50]
scenario 2.1, [2, 3, 10]->[6, 10]-> (6 + 10) [16]
scenario 2.2, [2, 3, 10]->[6, 10]-> (6 * 10) [60]
scenario 3.1, [2, 3, 10]->[12, 3]-> (12 + 3) [15]
scenario 3.2, [2, 3, 10]->[12, 3]-> (12 * 3) [36]
scenario 4.1, [2, 3, 10]->[20, 3]-> (20 + 3) [23]
scenario 4.2, [2, 3, 10]->[20, 3]-> (20 * 3) [60]
scenario 5.1, [2, 3, 10]->[13, 2]-> (13 + 2) [15]
scenario 5.2, [2, 3, 10]->[13, 2]-> (13 * 2) [26]
scenario 6.1, [2, 3, 10]->[30, 2]-> (30 + 2) [32]
scenario 6.2, [2, 3, 10]->[30, 2]-> (30 * 2) [60]

Summary: [2, 3, 10] -> (2 * 3 = 6) -> [6, 10] -> (6 + 10 = 16) -> [16]
```

### E.2   GPT-4 ATS-DFS ZERO-SHOT OUTPUT

```
Step 1
[1, 1, 7] -> (1 + 1 = 2) [2, 7]

Step 2
[1, 1, 7] -> [2, 7] -> (2 + 7 = 9)

This is not the goal. Let's step back. Now it is [2, 7].

Step 2 (revised)
[1, 1, 7] -> [2, 7] -> (2 * 7 = 14)

This is not the goal. Let's step back. Now it is [2, 7].

Let's step back. Now it is [1, 1, 7].

Step 1 (revised)
[1, 1, 7] -> (1 * 7 = 7) [7, 1]

Step 2
[1, 1, 7] -> [7, 1] -> (7 + 1 = 8)

Summary: [1, 1, 7] -> (1 * 7 = 7) -> [7, 1] -> (7 + 1 = 8) -> [8]
```

### E.3   GPT-4 CoT ZERO-SHOT OUTPUT

```
Step 1
[2, 3, 10] -> (2 * 3 = 6) -> [6, 10]

Step 2
[6, 10] -> (6 + 10 = 16) -> [16]

Summary
[2, 3, 10] -> (2 * 3 = 6) -> [6, 10] -> (6 + 10 = 16) -> [16]
```

### E.4   ToT-SOURCED TREE-TUNED DATA

```
# step = 0
The input is 2 3 10 16
The current states are:
[2, 3, 10]
From these states, we can achieve these new states:
[2, 3, 10] -> (2 + 3 = 5) -> [5, 10]\n", (value = 0)
```

```
[2, 3, 10] -> (2 * 3 = 6) -> [6, 10]\n", (value = 100)
[2, 3, 10] -> (10 / 3 = 3.3333333333333335) -> [2, 3.3333333333333335]\n
    ", (value = 23.3)
[2, 3, 10] -> (3 - 2 = 1) -> [1, 10]\n" (value = 0)
Then we choose the new states with largest values:
[2, 3, 10] -> (2 * 3 = 6) -> [6, 10]\n",
[2, 3, 10] -> (10 / 3 = 3.3333333333333335) -> [2, 3.3333333333333335]\n"

# step = 1
The input is 2 3 10 16
The current states are:
[2, 3, 10] -> (2 * 3 = 6) -> [6, 10]\n"
[2, 3, 10] -> (10 / 3 = 3.3333333333333335) -> [2, 3.3333333333333335]\n
From these states, we can achieve these new states:
[2, 3, 10] -> (2 * 3 = 6) -> [6, 10]\n [6, 10] -> (6 + 10 = 16) -> [16]\n
    ", (value = 100)
[2, 3, 10] -> (2 * 3 = 6) -> [6, 10]\n [6, 10] -> (6 * 10 = 60) -> [60]\n
    ", (value = 0)
[2, 3, 10] -> (2 * 3 = 6) -> [6, 10]\n [6, 10] -> (10 / 6 =
    1.6666666666666667) -> [6, 10]\n", (value = 100)
[2, 3, 10] -> (2 * 3 = 6) -> [6, 10]\n [6, 10] -> (10 - 6 = 4) -> [4]\n",
     (value = 33.3)
[2, 3, 10] -> (10 / 3 = 3.3333333333333335) -> [2, 3.3333333333333335]\n
    [2, 3.3333333333333335] -> (2 * 3.3333333333333335 =
    6.666666666666667) -> [6.666666666666667]\n", (value = 0)
[2, 3, 10] -> (10 / 3 = 3.3333333333333335) -> [2, 3.3333333333333335]\n
    [2, 3.3333333333333335] -> (2 + 3.3333333333333335 =
    5.333333333333334) -> [5.333333333333334]\n", (value = 33.3)
[2, 3, 10] -> (10 / 3 = 3.3333333333333335) -> [2, 3.3333333333333335]\n
    [2, 3.3333333333333335] -> (3.3333333333333335 - 2 =
    1.3333333333333335) -> [1.3333333333333335]\n" (value = 0)
Then we choose the new states with largest values:
[2, 3, 10] -> (2 * 3 = 6) -> [6, 10]\n [6, 10] -> (6 + 10 = 16) -> [16]\n
    ",
[2, 3, 10] -> (2 * 3 = 6) -> [6, 10]\n [6, 10] -> (10 / 6 =
    1.6666666666666667) -> [6, 10]\n"

# Summary
[2, 3, 10] -> (2 * 3 = 6) -> [6, 10]\n [6, 10] -> (6 + 10 = 16) -> [16]\n
    "
```