# OpenReview forum: "Autonomous Tree-search Ability of Large Language Models"
_ICLR.cc/2024/Conference — ICLR 2024 Conference Withdrawn Submission_

### Official Review · Reviewer_51yR · 2023-10-28

**Soundness:** 2 fair
**Presentation:** 2 fair
**Contribution:** 2 fair
**Rating:** 3
**Confidence:** 3

**Summary:**

The paper proposes developing "autonomous tree-search (ATS) ability" for large language models (LLMs) to allow them to solve reasoning tasks requiring exploration and search. ATS allows LLMs to generate responses demonstrating tree-structured search trajectories in either breadth-first search (BFS) or depth-first search (DFS) format. For large models like GPT-4, ATS ability can be activated through prompting the model to "role play" an assistant to generate the "tree" of the reasoning steps. Experiments on 4 puzzle tasks show GPT-4 with ATS prompts outperforms chain-of-thought. For smaller models like 7B/13B LLaMA, ATS ability can be acquired through supervised fine-tuning on GPT-4 generated ATS data. Experiments show ATS-tuned LLaMAs outperform both chain-of-thought tuned LLaMAs.

**Strengths:**

1. This paper proposes an approach to impart autonomous search ability to LLMs without specialized external programs. Could make LLMs more flexible and capable at complex reasoning. The ATS prompt approach seems quite generalizable to new tasks, requiring only fixed prompt. More flexible than task-specific passive search programs.
2. The empirical investigation covers both large and small LLMs. Demonstrates effectiveness over strong baselines like Tree of Thoughts on the puzzle solving tasks.

**Weaknesses:**

1. While the proposed idea presents a certain degree of improvement, its contribution appears incremental when compared to both CoT and ToT.
2. The scope of the literature survey is somewhat narrow. Notably absent are related works such as "PAL: Program-aided Language Models" and two concurrent studies that utilize pseudo-code-style prompts. These methodologies, too, emphasize simplicity and flexibility in generating reasoning steps.
3. Although the "Graph-of-thoughts" work is referenced, there is a lack of empirical comparison with it.
4. The empirical tasks primarily focus on simple games, which presents a limitation. It would be beneficial to incorporate more complex reasoning tasks. For instance, the ToT paper show the effectiveness of their methods through "Creative Writing."
5. LLM is known for its challenges in producing lengthy contextual answers, often leading to hallucinations. In contrast, ToT methods excel at breaking down complex tasks into more manageable steps, thereby enhancing the reliability of the output. The one-round prompting strategy, however, may encounter difficulties in handling intricate tasks, especially when they have significant breadth and depth.

**Questions:**

N/A

---

> ### Author Response · Authors · 2023-11-22
>
> Dear Reviewer 51yR,
>
> Thank you for your constructive comments. We have appropriately cited the works you mentioned and plan to running GoT as a baseline.
>
> According to your initial review, we have made the following efforts.
>
> - In **Additional Experiment 2**, a global response, we tested our methods on a real, complex reasoning task. Additionally, in conjunction with **Additional Experiment 1**, we demonstrated that ATS and ToT are not in conflict.
>
> Regarding hallucination, longer context typically suffers more severely. However, this also depends on whether the expected next generated token is within the LLM's capability.
>
> We claim that CoT is a method that can reduce hallucination through longer responses. Conversely, if we use prompt to turn off the CoT ability, the expected next generated token falls outside the LLM's capability, leading to hallucination even though GPT will have shorter response.
>
> The experimental accuracy show that ATS will not be seriously affected by hallucination in these tasks.
>
> Moreover, when it comes to significant depth and breath, we emphasize ATS and ToT are not in conflict as shown in **Additional Experiment 1**.
>
> Best,
>
> Authors

---

### Official Review · Reviewer_Tc6e · 2023-10-29

**Soundness:** 2 fair
**Presentation:** 3 good
**Contribution:** 2 fair
**Rating:** 5
**Confidence:** 4

**Summary:**

This paper studies using language models in tasks that benefit from systematic search. The paper proposes "Autonomous Tree Search", a prompting method that instructs the language model itself to perform tree search in-context, requiring a single call to the LLM. This contrasts to Tree of Thoughts, which uses an external program to guide search, repeatedly calling the LLM to propose next states and evaluate states. Experiments on 4 puzzles show improvements over CoT and ToT using GPT-4. Moreover, the authors explore distilling ATS from GPT-4 into smaller models (LLaMA 2 7B and 13B), showing that fine-tuning with ATS yields the best performance (e.g. compared to fine-tuning on GPT-4-generated ToT).

**Strengths:**

The paper proposes a simple idea that is very easy to apply in appropriate settings, so it's likely that it will be tried out by some of readers. The paper is well in scope around the current literature, and the idea itself is sound.

This is also the first work (that I'm aware of) that tried distilling tree-search into smaller models. An ongoing discussion in the area is whether there is a benefit of doing search in-context, since then one branch can be informed by others (in contrast to ToT, where the branches are expanded and evaluated independently). The experiments with LLaMA suggest that there might indeed by a noticeable difference between the two.

**Weaknesses:**

With regards to the method, one disadvantage that should be mentioned explicitly is that ATS is limited to search trees that fit in the context window. While this is not an issue for the simpler puzzles, more complex tasks might require extensive search that might not fit in-context. This limitation does not apply to Tree of Thoughts, so there is a trade-off between the two.

The main weaknesses of the paper in its current form are in the evaluation and presentation of the results.

First, the paper gives little insight into what drives the current results. While the idea of ATS is intuitive, it is not clear to me what factors allow it to have higher accuracy. While the authors show better overall accuracies than ToT, the lack of any qualitative analysis doesn't allow the reader to understand why that is the case. This is especially important when the evaluation is in tasks that themselves don't matter much, like puzzles. The main benefit of these simple, synthetic tasks is to allow us to get insights into the models. If not for that, higher accuracy on these tasks doesn't mean much by itself.

The "low-cost setting" of Tree of Thoughts does not seem to be tree search at all. If the search width is 1, then the model is ultimately only following a single path, even if at each level it will propose multiple next steps. The fact that this performs even worse that CoT (which I can speculate, but don't fully understand from the paper) indicates to me that this comparison is unfair.

The cost evaluation in cents will get old fast. I'd suggest maybe showing some of them in the paper as with a note that "at the time of writing" these are the costs, but these numbers might be meaningless for readers even a few months from now.

For fine-tuning, it would have been useful to also compare to fine-tuning on CoT generated by the larger model, as done in recent prior work.

There seems to be quite a bit of redundancy between the figures and tables (e.g., the cost/accuracies in Figures 3 and 4, then in Table 1). A lot of this space could have been used to give insight into some of the numbers.

Finally, and perhaps the main weakness: I found the choice of tasks a bit arbitrary. The authors propose 4 puzzles and only evaluate on those, without reference to prior literature. The paper makes a point about the choice of tasks in the Tree of Thoughts paper (that they don't fully isolate the tree search ability), but I don't think that that justifies discarding them completely.

**Questions:**

- Why would "low-cost ToT" perform worse than CoT in some settings (noticeably so in the Drop Water puzzle)?
- What are the main failure modes you observed for ToT, and why does ATS seem to address them?
- Are there any viable applications of ATS in existing tasks that are not synthetic puzzles?
- Why is the output cost for ATS often smaller in the 4-shot setting, compared to 0-shot?
- Why do figures 2 and 3 not include the low-cost results for other methods other than ToT?

---

> ### Author Response · Authors · 2023-11-22
>
> Dear Reviewer Tc6e,
>
> Thanks again for your detailed, helpful, and constructive comments. We added "at the time of writing". We plan to finetune on CoT generated by the larger model and rearrange our figures and tables.
>
> In response to your initial review, we have made the following efforts:
>
> - We made **Additional Experiment 2** in global response, we tested our methods in a real complex reasoning task **CrossWords**, which is one of the tasks from ToT paper. This task requires strong understanding of rows and columns, making all in-context methods difficult in solve the puzzle.
> - We provided an **Additional Analysis**. The width = 1 setting is mentioned in original ToT paper. We explained the reasons behind ToT's failure on the Drop Water Puzzle.
>
> **Q1.Q2. ToT performance.** We anticipate that low-cost ToT will perform similarly to CoT in most cases. The exception, the Drop Water Puzzle, is discussed in the **Additional Analysis** below.
>
> **Q3. non synthetic puzzles.** **Additional Experiment 2** in global response shows the results and **Additional Experiment 1** shows another applications of ATS to combined with ToT.
>
> **Q4. few-shot response shorter.** Few-shot responses contain task-specific knowledge that helps prune some unnecessary branches. For example, in the Drop Water Puzzle, it is meaningless to fill the second bottle immediately after filling the first bottle.
>
> **Q5. figure.** The high-cost setting for in-context methods is only self-consistency, resulting in marginal improvements. We selected key information from the table and aimed to make the figure more concise.
>
> Best,
>
> Authors

---

> > ### Author Response · Authors · 2023-11-22
> > **Additional Analysis**
> >
> > This section explains the unsatisfactory result of ToT on the Drop Water Puzzle and how in-context methods avoid failure.
> >
> > **ToT fails on Drop Water Puzzle**
> >
> > Since the Drop Water Puzzle requires finding a solution with a restriction on the number of steps, we use the metric `estimating the minimum future steps` as evaluator in ToT.
> >
> > Here is an instance of the puzzle.
> >
> > ```
> > Two bottles with volumes of 10 and 16.
> > Find a solution to result in 6 within 2 steps.
> > ```
> >
> > The only solution is `[0 / 10, 0 / 16] -> [0 / 10, 16 / 16] -> [10 / 10, 6 / 16]`.
> >
> > For ToT$_{\text{width}=1}$ setting, it only remains one branch, hence comparing `[10 / 10, 0 / 16]` and `[0 / 10, 16 / 16]` almost determines whether ToT solve it.
> >
> > However, evaluating a state is difficult for GPT-4.
> > - Evaluate `[10 / 10, 0 / 16]`. GPT's response : `Step = 1`
> > - Evaluate `[0 / 10, 16 / 16]`. GPT's response : `Step = 3` or `Step = 4`
> >
> > Even with the help of few-shot, `[10 / 10, 0 / 16]` wins `[0 / 10, 16 / 16]`. We believe the major reason is that **evaluating a state in Drop Water Puzzle is very confusing**. The whole prompt is shown below, calling gpt directly through a single user query `[{"role": "user", "content":prompt}]`.
> >
> > ```
> > # Game Description
> > This game involves two empty bottles with capacities of 10 and 16 liters. The objective is to collect exactly 6 liters of water, using a large water reservoir.
> >
> > Each step of the game offers you the choice to execute one of two actions:
> >
> > Opt for a bottle and proceed to fill it to its brim from the reservoir or conversely, empty it completely back into the reservoir. The following are some illustrative examples:
> > [0 / 7, 7 / 22] -> (fill the first) [7 / 7, 7 / 22];
> > [8 / 8, 11 / 19] -> (empty the first) [0 / 8, 11 / 19].
> >
> > Pour water from one bottle into the other. This action concludes once the recipient bottle is at its full capacity or the source bottle is entirely empty. Here are some practical examples:
> > [0 / 7, 9 / 9] -> (pour the second to the first, it will halt by the full recipient) [7 / 7, 2 / 9];
> > [5 / 5, 5 / 7] -> (pour the first to the second, it will halt by the full recipient) [3 / 5, 7 / 7];
> > [12 / 12, 0 / 17] -> (pour the first to the second, it will halt by the empty source) [0 / 12, 12 / 17];
> >
> > # Your Task
> > Now you are facing a subtask of this game. Given a state, estimate how many steps to reach the goal 6.
> >
> > Q: [8 / 8, 11 / 19] to get 11
> > A: Steps = 0
> > Q: [10 / 10, 10 / 29] to get 20
> > A: Steps = 1
> > Q: [0 / 20, 0 / 25] get 15
> > A: Steps = 4
> > Q: [0 / 11, 13 / 13] get 2
> > A: Steps = 1
> > Q: [0 / 10, 16 / 16] to get 6
> > A:
> > ```
> >
> > **In-context methods**
> >
> > Conversely, in the CoT setting, especially in the few-shot setting, it is within the capability of GPT-4 to learn **the pattern "fill larger bottle and then pour to the smaller one"**. We believe **understanding this pattern is easier than understanding how to evaluate a state in this puzzle**.
> >
> > ```
> > Step 1
> > [0 / 10, 0 / 16] -> (fill the second) [0 / 10, 16 / 16]
> >
> > Step 2
> > [0 / 10, 16 / 16] -> (pour the second to the first) [10 / 10, 6 / 16]
> > ```
> >
> > **Further investigation**
> >
> > To further figure out this phenomenon, we apply ATS to the State Evaluator of ToT in Additional Experiment 1. The result shows that the issue is related to the difficulty in evaluation.

---

> > > ### Comment · Reviewer_Tc6e · 2023-11-22
> > > **Response**
> > >
> > > Thank you for the detailed response. The clarifications mostly make sense to me. The experiment using ATS as an evaluation is an interesting direction as well.
> > >
> > > I appreciate the effort to run on Crosswords. Unfortunately, I think the results highlight a limitation of a method that solely relies on the LLM to do everything in context, which is that it doesn't offer a way to circumvent limitations of the underlying LLM. Thus, for example, if the LLM cannot read rows/columns properly, then it won't be able to do that while doing search in context either.
> > >
> > > ATS does have a clear cost advantage compared to ToT, which gets expensive fast. I strongly suggest the authors find more realistic tasks where that advantage really shines. Those would need to be tasks where the LLM can execute all intermediate operations reliably but perhaps would fail at planning with simple CoT.
> > >
> > > Thus I'd like to keep my score.

---

### Official Review · Reviewer_Mb1G · 2023-10-30

**Soundness:** 3 good
**Presentation:** 3 good
**Contribution:** 2 fair
**Rating:** 5
**Confidence:** 4

**Summary:**

The paper proposes developing "autonomous tree-search ability" in large language models (LLMs) to enhance their reasoning and problem-solving capabilities. In contrast with previous work like chain-of-thought and tree-of-thought, this paper demonstrates a prompting method to help large language models to automatically conduct planning without external planners (such as tree-search). Experiments with GPT4 over four puzzle games verify the effectiveness of ATS compared with CoT and ToT. By collecting ATS data from GPT4, the authors successfully show a much smaller model (such as LLaMA-7b/13b) can be supervise-finetuned to have the similar ability.

**Strengths:**

The paper has the following strengths:

1. The paper writing is overall clear and straightforward.
2. The idea is overall novel.
3. The author conducts the experiment by prompting GPT4 and training on smaller LLaMA models to validate the general capability of such methods.

**Weaknesses:**

The paper has the following Weaknesses:

1. The experiments are not comprehensive from a few perspectives.
1.1 The evaluation is limited to just 4 puzzle games. More complex reasoning tasks should be tested to better validate the value of autonomous tree search.
1.2 The author mainly analyzes the performance and the paper lacks in-depth analysis. For example, there is no analysis of how search spaces, branching factors, solution depth, search algorithms, and different prompt variations can affect performance.

2. The author is supposed to discuss more about the limitations of ATS. For instance, ATS seems to generate more tokens and is more easily constrained by the model context length. Also, It seems hard for ATS to generalize to complex and long-term planning problem.  I recommend the authors include several limitation discussions like this (experiments will be appreciated).

3. Lack of enough related work. A lot of work is discussing how to combine tree search with LLM, for example (Hao et al., 2023) and I am sure there have been more in the past few years. The author should add these literatures as related work.


Reference
Hao, Shibo, et al. "Reasoning with language model is planning with world model." arXiv preprint arXiv:2305.14992 (2023).

**Questions:**

1. Will GPT4 generate wrong planning during the ATS process? How do you filter the wrong planning out from the dataset?
2. You mentioned that DFS exploration was inconsistent without task-specific examples. Did you investigate why DFS was not as robust? Are there ways to improve the generalizability of DFS search?

---

> ### Author Response · Authors · 2023-11-22
>
> Dear Reviewer Mb1G,
>
> Thanks again for your constructive comments. We have cited the work you've mentioned.
>
> Following your initial review, we have implemented the following improvement:
>
> - In **Additional Experiment 2** involving a global response, we tested our methods in a real, complex reasoning task. The results highlight ATS's effectiveness among in-context methods.
> - With **Additional Experiment 2**, we discussed our limitations. All in-context methods suffer from the limitations of LLMs, for example the understanding of rows and columns.
> - Additionally, in **Additional Experiment 1**, we demonstrated that ATS and ToT are not in conflict, indicating some limitations would be addressed by combining them.
>
> **Q1. filter the wrong planning.** In the problem definition, we introduced a summary section at the end of the response. For example, the summary might resemble `[0 / 10, 0 / 16] -> [0 / 10, 16 / 16] -> [10 / 10, 6 / 16]`. This summary allows us to confirm the correctness of the solution path, disregarding other paths.
>
> **Q2. DFS not robust.** One reason is that **pruning process** in DFS is usually **not robust**. In the ToT repository, DFS code is provided for crossword tasks. I have tried to reproduce the experiment and found **the pruning process is even sensitivity to version changes of GPT-4**.
>
> Similar challenges are encountered in our situation. Deciding whether to prune is difficult. There is no common principal of pruning. That's why **task-specific knowledge** from few-shot learning significantly aids in this decision-making process and contributes to stabilize DFS method.
>
> Conversely, for the BFS method, LLMs only need to compare states on the same level, which is more stable than deciding a branch to prune or not with only the information of this branch.
>
> Best,
>
> Authors

---

### Official Review · Reviewer_Dq3i · 2023-10-31

**Soundness:** 3 good
**Presentation:** 3 good
**Contribution:** 2 fair
**Rating:** 6
**Confidence:** 3

**Summary:**

This paper presents a prompting method to make LLM automatically perform tree search (such as BFS and DFS) in one text completion. In the 4 synthetic puzzles tested, the method demonstrate gains over Tree of Thought, especially under zero-shot low cost setting. The proposed method can also be used to collect reasoning traces, which is then used to fine-tune smaller LMs to improve its performance.

**Strengths:**

- The paper is mostly well-written and easy to follow
- The proposed method is simple and novel, although can be seen as one type of chain of thought.
- The experiment results demonstrate substantial gains over CoT and ToT.

**Weaknesses:**

- The experiment settings are very toy/synthetic, making it unclear whether the method is useful for more realistic tasks.

**Questions:**

n/a

---

> ### Author Response · Authors · 2023-11-22
>
> Dear Reviewer Dq3i,
>
> Thank you once again for your constructive comments.
>
> We are greatly encouraged by your positive feedback on the quality of our writing and the novelty of our ideas.
>
> In response to your initial review, we conducted **Additional Experiment 2** on a real, complex reasoning task with a global scope. The results indicate that ATS is effective among in-context methods. ATS handles real complex reasoning tasks better than CoT.
>
> Best,
>
> Authors

---

> > ### Comment · Reviewer_Dq3i · 2023-11-22
> >
> > Thank you for the detailed response. I will maintain the current score.

---

### Author Response · Authors · 2023-11-22
**Global Response**

Dear AC and all reviewers,

We sincerely thank all the reviewers for their insightful feedbacks! We are encouraged that they found the paper is well in scope around the current literature (Tc6e), the paper is mostly well-written (Dq3i) and clear (Mb1G), the idea is novel (Dq3i, Mb1G), simple (Mb1G, Tc6e), and sound (Tc6e). We are also encouraged by the reviewers' appreciation of our effort on investigating both large and small LLMs (Mb1G, 51yR).

Furthermore, we value the reviewers' insightful and constructive suggestions and concerns. In this general response, we emphasize our main contributions and provide a summary of the experiments added.

**Contributions.**

- **In-context and Flexibility.**
    - We appreciate reviewer Tc6e for emphasizing the ongoing discussion in the area regarding the potential benefits of conducting searches in-context. This serves as the main motivation driving our commitment to being pioneers in investigating in-context search abilities and distilling tree-search into smaller models.
    - We want to clarify that our goal is **not** to develop a **system** with LLMs. Instead, we aim to demonstrate the capability to address search issues **solely through LLMs**. The flexibility of ATS not only implies the convenience of solving various problems but also operates at a higher scope. Even if a perfect system with LLMs exists, the ongoing development of in-context methods remains necessary. For instance, in-context ability, as opposed to a system, enables further tuning/RLHF/providing instructions.
    - Furthermore, the in-context method and a system with LLMs are not in conflict. ATS can be combined with all systems utilizing LLMs, enhancing their performance. For example, it can serve as the evaluating function for Tree-of-Thoughts *(Additional Experiment 1)*.

In Short,

- Proposing flexible and capable methods to unlock or enhance tree-search abilities for both large and small LLMs.
- Conducting experiments on four puzzles, demonstrating the significant efficiency of ATS, which calls LLM only once. Additionally, showcasing its effectiveness in a real complex reasoning task *(Additional Experiment 2)* attached.

**Additional Experiments.**

We have added an additional experiments section to the global response.

- Additional Experiment 1. We combined ATS with ToT, emphasizing they are not in conflict. For tasks requiring complex evaluation, ATS proves capable of improving ToT.
- Additional Experiment 2. We tested ATS on a real complex reasoning task. Despite the inherent limitations of manipulating rows and columns in all in-context methods, ATS outperformed other in-context methods.

---

> ### Author Response · Authors · 2023-11-22
> **Additional Experiment 1**
>
> ToT + ATS. Incorporate ATS into State Evaluator of ToT.
>
> - The results of two tasks are shown below, indicating **ATS technique can enhance ToT by strengthening evaluator**.
>
> |                                          | Drop Water | Number Path |
> | ---------------------------------------- | ---------- | ----------- |
> | ToT $_{\text{width}=1}$ **without** ATS-evaluator (few-shot) | 14.8       | 52.8        |
> | ToT $_{\text{width}=1}$ **with** ATS-evaluator (few-shot)    | 73.2       | 90.6        |
>
> **Insights of Additional Experiment 1**
>
> - Drop Water Puzzle
>     - We use an Evaluator to estimate the minimum future steps. Fewer future steps mean a better state.
>     - An Evaluator with ATS can accurately estimate the minimum future steps. For instance, the example below shows GPT-4 generates `[0 / 11, 13 / 13] -> (pour 2 to 1) [11 / 11, 2 / 13]` and determines answer `Steps = 1`.
> ```
> # Prompt
> (In this task, you need to estimate the minimum future steps.)
> (Problem Definition and some shots)
> Q: [0 / 10, 16 / 16] to get 6
> A:
> ------------------------------------------------------------
> # Response
> Start search.
> Step 1:
> Scenario 1 [0 / 10, 16 / 16] -> (pour 2 to 1) [10 / 10, 6 / 16] (Detect 6, Finish!)
> Summary: Steps = 1
> ```
>
> - Number Path Puzzle
>     - We use an Evaluator to estimate the probability of finding a solution from the current state. A higher probability means a better state.
>     - An Evaluator with ATS can accurately estimate the minimum future steps. For instance, the example below shows GPT-4 generates several search trials and determines the answer `Probability = 0`.
>
> ```
> # Prompt
> (In this task, you need to estimate the probability to find a solution from current state.)
> (Problem Definition and some shots)
> Q: 5 to 12 with exactly 2 operation.
> A:
> ------------------------------------------------------------
> # Response
> Begin search.
> Step 1:
> scenario 1. 5 -> (+1) -> 6
> scenario 2. 5 -> (*2) -> 10
> Step 2:
> scenario 1.1. 6 -> (+1) -> 7 (Discard. Because 7<13.)
> scenario 1.2. 6 -> (*2) -> 12 (Discard. Because 12<13.)
> scenario 2.1. 10 -> (+1) -> 11 (Discard. Because 11<13.)
> scenario 2.2. 10 -> (*2) -> 20 (Discard. Because 20>13.)
> Summary: Probability = 0
> ```

---

> ### Author Response · Authors · 2023-11-22
> **Additional Experiment 2**
>
> A realistic and challenging reasoning task: **CrossWords**
>
> **Task Definition:** Solve 5x5 mini crosswords. Given 5 horizontal clues and 5 vertical clues as input, generate thoughts about which 5-letter word fits each clue. The output should consist of 5 rows, where each row contains a 5-letter word separated by space.
>
> - The results of this task are presented below. The `IO`, `CoT`, `ToT` results refer to the ToT paper. The three metrics (`letter`, `word`, `game`) represent accuracy over letters, words, and games, respectively. The results indicate:
>     - 1) **ATS handles real complex reasoning tasks better than CoT.**
>     - 2) All in-context methods have similar orders of magnitude of cost, while ToT incurs much higher costs.
>     - 3) **One major limitation of ATS is its constraint on the ability of LLMs.** This task requires a strong understanding of rows and columns, and GPT-4 often fails in this aspect. (some examples below)
> Conversely, ToT decomposes some of the difficulty of rows and columns through Python code. (*e.g.,* writing back to the table with a row/column, extracting a row/column to a flat style before evaluation)
>
>
> |                | Letter(%) | Word(%) | Game(%) | prompt tokens | generate tokens |
> | -------------- | ------ | ---- | ---- | ---- | ---- |
> | IO (few-shot) | 38.7   | 14   | 0    | 790.25 | 30.505
> | CoT (few-shot) | 40.6   | 15.6 | 1    | 1448.25 | 162.81 |
> | ToT (few-shot) | 78     | 60   | 20   | >584306.45 | >848.05
> | ATS-BFS (one-shot) | 46.6   | 18.5 | 0 | 1549.25 | 1211.75
>
> (According to https://github.com/princeton-nlp/tree-of-thought-llm/tree/master/logs/crosswords, `infoss_dfs_prune.json` shows that there are 16961 evaluations in total and each evaluation has at least 689 tokens. 16961*689/20=584306.45)
>
> **GPT-4 often fails in manipulating rows and columns**
>
> Here are two examples we met that GPT-4 fails in in understanding rows and columns.
>
> ```
> _ K _ _ D
> _ A _ _ R
> _ U _ _ O
> _ R _ _ N
> _ I _ _ Y
>
> Step 2.
> Fill row 1 with SKALD,
> and fill row 2 with WATER.
> Then the matrix is:
> S K A L D
> W A _ _ R
> _ U _ _ O
> _ R _ _ N
> _ I _ _ Y
> ```
>
> ```
> scenario 1.1
> S A W E R
> A G A R I
> R A T E R
> M E A D S
> H E A R D
> h1. match SAWER with 5 letters
> ...
> h5. match HEARD with 5 letters
> v1. match SURGE with 3 letters
> ```